# Ischemic stroke in patients that recover from COVID-19: Comparisons to historical stroke prior to COVID-19 or stroke in patients with active COVID-19 infection

Naveed Akhtar[1], Fatma Abid[2], Rajvir Singh[3], Saadat Kamran[1], Yahia Imam[1], Salman Al-Jerdi[4], Sarah Salamah[2], Rand Al Attar[2], Muhammad Yasir[5], Hammad Shabir[6], Deborah Morgan[1], Sujatha Joseph[1], Muna AlMaslamani[2], Ashfaq Shuaib[7] *

1 The Neuroscience Institute, Hamad Medical Corporation, Doha, Qatar, 2 Infectious Diseases Section, Medicine Department, Hamad Medical Corporation, Doha, Qatar, 3 Cardiology Research, Hamad Medical Corporation, Doha, Qatar, 4 Weill Cornell Medical College, Doha, Qatar, 5 Emergency Department, Hamad Medical Corporation, Doha, Qatar, 6 Medicine Department, Hamad Medical Corporation, Doha, Qatar, 7 University of Alberta, Edmonton, Alberta, Canada

* ashfaq.shuaib@ualberta.ca

**Data Availability Statement:** All relevant data are within the paper.

## Abstract

### Background and purpose

Understanding the relationship of COVID-19 to stroke is important. We compare characteristics of pre-pandemic historical stroke (Pre-C), cases in acute COVID infection (Active-C) and in patients who have recovered from COVID-19 infection (Post-C).

### Methods

We interrogated the Qatar stroke database for all stroke admissions between Jan 2019 and Feb 2020 (Pre-C) to Active-C (Feb2020-Feb2021) and Post-C to determine how COVID-19 affected ischemic stroke sub-types, clinical course, and outcomes prior to, during and post-pandemic peak. We used the modified Rankin Scale (mRS) to measure outcome at 90-days (mRS 0–2 good recovery and mRS 3–6 as poor recovery). For the current analysis, we compared the clinical features and prognosis in patients with confirmed acute ischemic stroke.

### Results

There were 1413 cases admitted (pre-pandemic: 1324, stroke in COVID-19: 46 and recovered COVID-19 stroke: 43). Patients with Active-C were significantly younger, had more severe symptoms, fever on presentation, more ICU admissions and poor stroke recovery at discharge when compared to Pre-C and Post-C. Large vessel disease and cardioembolic disease was significantly more frequent in Active-C compared to PRE-C or post-C.

### Conclusions

Stroke in Post-C has characteristics similar to Pre-C with no evidence of lasting effects of the virus on the short-term. However, Active-C is a more serious disease and tends to be more severe and have a poor prognosis.

**Funding:** The authors received no specific funding for this work.

**Competing interests:** The authors have declared that no competing interests exist.

**Abbreviations:** Pre-C, pre-pandemic stroke; Active-C, COVID infection related stroke; Post-C, recovered from COVID-19 infection.

## Introduction

The number of COVID-19 cases worldwide exceeded 430,000,000 as of February 24, 2022 with most patients recovering from the infection [1]. COVID-19 may affect the cardiovascular system and increases the risk of venous thrombosis and pulmonary embolism [2], myocardial injury [3] and stroke [4, 5]. Acute stroke has been reported in 0.5–2.5% of active COVID-19 and tends to be more severe with a higher mortality [4]. Stroke in active COVID-19 tends to be severe and caries a higher rate of mortality [4–7]. To our knowledge, there are no studies that have compared the stroke phenotype in patients that recover from COVID-19 infection to stroke in patients with no previous history of stroke or when stroke occurs following a complete recovery from the infection.

We have previously published on acute stroke in COVID-19 pandemic from Qatar [7, 8]. Our previous research compared the rates of stroke admissions prior to the pandemic and during the pandemic. There was a significant increase in patients with severe stroke and there were fewer patients admitted with diagnosis of 'stroke mimic' during the first phase of the COVID-19 pandemic. Our main objective for the current research was to compare the clinical presentation, severity and type of stroke, and prognosis in strokes that were admitted to our hospital prior to the COVID-19 pandemic (Pre-C), stroke in patients with active COVID-19 infection (Active-C) and the development of stroke in subjects who had full clinical recovery from COVID-19 (Post-C).

## Methods

The Qatar Stroke Database prospectively collects information on most acute stroke (98%) admitted in Qatar to the Hamad General Hospital (HGH) since February 2013 as previously published [9, 10]. The Institutional Review Board, Hamad Medical Corporation at the Medical Research Centre (MRC-01-20-489) approved the study. Data will be made available on request.

All acute stroke patients admitted to HGH between January-2019 to February-2020 were evaluated for the study (Pre-C) and served as the reference comparator for the COVID-19 cases. The patients who developed stroke while they had active COVID-19 positive were admitted to the hospital between February 2020 (when the first cases of COVID-19 were reported in Qatar) and February 2021. The active-C cases all had active viral disease at the time of the stroke and the Post-C patients had all recovered from the viral illness at the time of the stroke. All patients in the Post-C group had a confirmation of the diagnosis of COVID-19 infection with rt-PCR testing. All patients in post-C group were tested for COVID-19 had fully recovered from the viral infection and no patients displayed symptoms associated with long-COVID. The clinical information including risk factors, investigations, clinical presentation, and course during hospitalization were recorded. The severity of symptoms at admission (NIHSS score), clinical diagnosis as defined by the TOAST classification [11] and Bamford classification [12], and the length of stay in hospital are also recorded. The modified Rankin Scale (mRS) pre-admission, at discharge, and at 90-day follow-up are also documented.

### Patient and public involvement

Patients or the public WERE NOT involved in the design, or conduct, or reporting, or dissemination plans of our research.

### Statistical analysis

Descriptive statistics in the form of mean and standard deviations for continuous variables and frequency with percentages for categorical variables were performed. One-way ANOVAs

with post hoc (Bonferroni) analyses were performed to see significant mean level differences for all continuous variables according to Pre-COVID, Active COVID and Post-COVID stroke groups. Chi-Square tests with standardized residuals were calculated to see association with categorical variables and the groups. Multivariate logistic regression analysis was performed to see associated risk factors to 90 days poor outcome. Adjusted odds ratio (OR) with 95% C.I. and P values were presented. P value less than equal to 0.05 (two tailed) was considered statistically significant level. SPSS 28.0 statistical package was used for the analysis.

## Results

There were 1413 patients [age; 54.2 ± 12.9 male 1156/1413 (81.8%) female 257/1413 (18.2%)] admitted to HGH during the study period and available for analysis. Of the 1413 stroke patients, there were 1324 patients admitted without COVID-19 in the 14 months prior to the pandemic (Pre-C), 46 cases with active COVID-19 infection (Active-C) and 43 COVID-19-recovered cases (Post-C) as shown in the Table 1.

There was no significant difference in the age of the three groups. The higher percentage of males reflects the demographics of Qatar with a predominantly male expatriate population as have been previously reported [9, 10]. The mean duration of time between recovery from COVID-19 infection and stroke was 126.9±75.9 days (median 124 days). Small vessel disease (SVD) is the most common type of stroke in the Qatari and expatriate population, likely due to the high prevalence of poorly controlled hypertension and diabetes as has been previously documented [10]. SVD was significantly lower in active-C (10.9%) compared to 45.7% in pre-C and 27.9% in post-C (p<0.001). The active-C group was associated with an increase in the percentage of large vessel and embolic stroke as shown in the Table 1.

The active-C patients were more likely to have higher NIHSS on admission and significantly more patients had cortical strokes. The admission NIHSS was 10.8± 8.6 in active-C, compared to 4.8± 5.9 in the pre-C and 6.1±7.4 in post-C patients (p <0.001). Active COVID-19 patients were more likely to be febrile (28.3% versus pre-C (0.5%) and post-C (9.3%). Patients with active COVID-19 and stroke were more likely to have admissions to the ICU (active-C: 32.6%, pre-C: 4.8% and post-C: 14%; p<0.001), more frequently required intubation (active-C:19.6%, pre-C: 4.61% and post-C: 7%; p<0.001) and had longer length of hospitalization [LOC] (active-C: 29.1±31.0, pre-C: 5.3±5.5 and post-C: 11.5±29.1 days, P<0.001). One Way ANOVA with Bonferroni post-hoc analysis was performed to see statistical significance of mean differences between the groups of NIHSS on admission and length of stay. On Post-hoc analysis of NIHSS on admission, active-C patients score was significantly higher than the pre-C patients, whereas there was no statistical difference between pre-C and post-C patients NIHSS score on admission, (p<0.001). On Post-hoc analysis of length of stay, active-C and post-C patient's duration of stay was significantly prolonged when compared to the pre-C patients (p<0.001).

Patients with stroke following recovery from COVID-19 (post-C) had a clinical profile very similar to pre-C patients. These patients had all fully recovered from the acute infection, and none had any symptoms suggestive of profiles of COVID-19 long-haulers. They had milder neurological disease on admission, and similar mRS at discharge (mRS [0–2] active-C: 23.9%, pre-C: 59.7% and post-C: 62.8% P <0.001). They were, however, more likely to be febrile on admission compared to pre-C. Recovering COVID-19 stroke patients with fever had similar clinical course and prognosis to patients without fever (poor outcome [mRS 3–6] 20.0 vs 21.6%, p = 0.93) and all febrile recovering stroke patients had no evidence of active COVID-19 infection.

**Table 1. Demographic and clinical characteristics of patients with recovered COVID-19, active COVID-19, and pre-pandemic stroke patients.**

| Characteristics or Investigations | Total Stroke Cases (n = 1413) | Pre-COVID Stroke (n = 1324) | Active-COVID Stroke (n = 46) | Post-COVID Stroke (n = 43) | P Value |
|---|---|---|---|---|---|
| Age, Mean, years | 54.2 ±12.9 | 54.3 ±12.9 | 51.3 ±10.2 | 54.4 ±14.3 | 0.31 |
| Sex—Male | 1156 (81.8) | 1077 (81.3) | 44 (95.7) | 35 (81.4) | 0.05 |
| Female | 257 (18.2) | 247 (18.7) | 2 (4.3) | 8 (18.6) | |
| **Risk factors** | | | | | |
| Hypertension | 1033 (73.1) | 986 (74.5) | 21 (45.7) | 26 (60.5) | <0.001 |
| Diabetes | 775 (54.8) | 732 (55.3) | 19 (41.3) | 24 (55.8) | 0.17 |
| Dyslipidemia | 770 (54.5) | 754 (56.9) | 3 (6.5) | 13 (30.2) | <0.001 |
| Atrial Fibrillation on Admission | 75 (5.3) | 70 (5.3) | 0 | 5 (6.7) | 0.05 |
| Active Smoking | 456 (32.3) | 439 (33.2) | 8 (17.4) | 9 (20.9) | 0.02 |
| Prior Stroke | 171 (12.1) | 163 (12.3) | 4 (8.7) | 4 (9.3) | 0.001 |
| Coronary Artery Disease | 183 (13.0) | 174 (13.1) | 4 (8.7) | 5 (11.6) | 0.65 |
| BMI on admission (mean) | 27.8 ±5.1 | 27.9 ±5.1 | 26.2 ±4.0 | 26.5 ±4.4 | 0.01 |
| Fever on Admission | 24 (1.7) | 7 (0.5) | 13 (28.3) | 4 (9.3) | <0.001 |
| NIHSS on admission (mean) | 5.1 ±6.1 | 4.8 ±5.9 | 10.8 ±8.6 | 6.1 ±7.4 | <0.001 |
| **NIHSS Severity** | | | | | |
| Mild (NIHSS 0–4) | 939 (66.5) | 898 (67.8) | 14 (30.4) | 27 (62.8) | <0.001 |
| Moderate (NIHSS 5–10) | 270 (19.1) | 248 (18.7) | 14 (30.4) | 8 (18.6) | |
| Severe (NIHSS >10) | 204 (14.4) | 178 (13.4) | 18 (39.1) | 8 (18.6) | |
| IV Thrombolysis given | 146 (10.3) | 140 (10.6) | 4 (8.7) | 2 (4.7) | 0.42 |
| Thrombectomy done | 74 (5.2) | 70 (5.3) | 1 (2.2) | 3 (7.0) | 0.57 |
| ICU Admission | 85 (6.0) | 64 (4.8) | 15 (32.6) | 6 (14.0) | <0.001 |
| Intubated during Admission | 73 (5.2) | 61 (4.6) | 9 (19.6) | 3 (7.0) | <0.001 |
| **TOAST Classification** | | | | | |
| Small Vessel Disease | 622 (44.0) | 605 (45.7) | 5 (10.9) | 12 (27.9) | <0.001 |
| Large Vessel Disease | 231 (16.3) | 210 (15.9) | 13 (28.3) | 8 (18.6) | |
| Cardioembolic | 359 (25.4) | 329 (24.8) | 16 (34.8) | 14 (32.6) | |
| Stroke of Determined Origin | 86 (6.1) | 75 (5.7) | 6 (13.0) | 5 (11.6) | |
| Stroke of Undetermined Origin | 115 (8.1) | 105 (7.9) | 6 (13.0) | 4 (9.3) | |
| **Prognosis at Discharge** | | | | | |
| Good (mRS 0–2) | 829 (58.7) | 791 (59.7) | 11 (23.9) | 27 (62.8) | <0.001 |
| Poor (mRS 3–6) | 584 (41.3) | 533 (40.3) | 35 (76.1) | 16 (37.2) | |
| **Prognosis at 90-Days (n = 1088)** | | | | | |
| Good (mRS 0–2) | 721 (66.3) | 671 (67.2) | 18 (39.1) | 32 (74.4) | <0.001 |
| Poor (mRS 3–6) | 367 (33.7) | 328 (32.8) | 28 (60.9) | 11 (25.6) | |
| Mortality at Discharge | 19 (1.3) | 18 (1.4) | 1 (2.2) | 0 | 0.66 |
| Mortality at 90-Days (n = 1088) | 49 (4.5) | 42 (4.2) | 5 (10.9) | 2 (4.7) | 0.10 |
| Characteristics or Investigations | Total Stroke Cases (n = 1413) | Pre-COVID Stroke (n = 1324) | Active-COVID Stroke (n = 46) | Post-COVID Stroke (n = 43) | P Value |
| Heart rate | 81.8 ±14.9 | 81.6 ±14.9 | 84.8 ±13.8 | 84.7 ±13.7 | 0.16 |
| Systolic Blood Pressure | 155.8 ±30.5 | 156.4 ±30.6 | 147.6 ±33.3 | 146.7 ±22.5 | 0.02 |
| Diastolic Blood Pressure | 90.6 ±19.2 | 90.6 ±19.3 | 88.6 ±20.1 | 92.5 ±15.3 | 0.63 |
| Platelet Counts | 270.3 ±78.5 | 268.7 ±75.3 | 302.6 ±123.9 | 285.5 ±101.1 | 0.007 |
| HbA1c on Admission | 7.6 ±4.8 | 7.5 ±4.9 | 7.9 ±2.5 | 7.7 ±2.4 | 0.87 |
| Length of Stay | 6.3 ±10.1 | 5.3 ±5.5 | 29.1 ±31.0 | 11.5 ±29.1 | <0.001 |

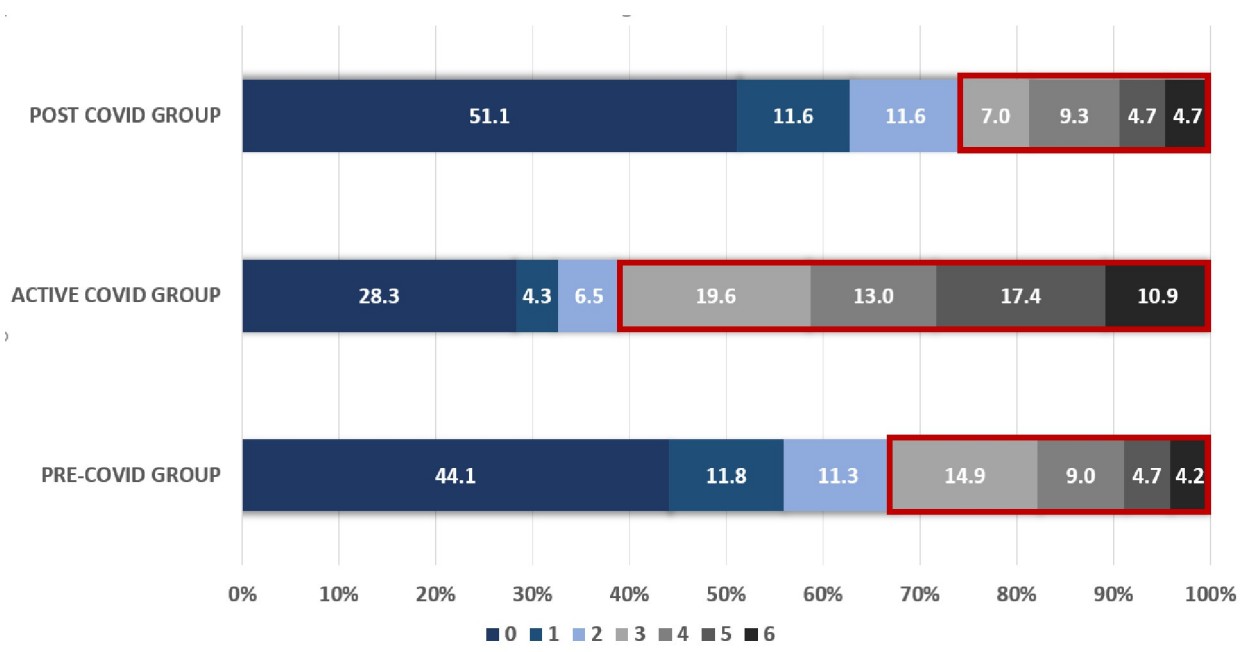

**Fig 1. Showing outcome of patients with active COVID-19 infection have significantly fewer patients who improved to a mRS of 0–2 at the 90-days follow-up compared to patients who never had the viral infection or who fully recovered from the infection.**

Patients in whom acute stroke occurred during active COVID-19 infection had slower recovery during hospitalization and at 90-days follow-up when compared to pre-C or post-C patient as shown in Fig 1. At 90-day follow up, good recovery (mRS 0–2) was seen in 39.1% in active-C patients compared to 67.2% in pre-C and 74.4% in post-C subjects (P <0.001).

Adjusting age and sex in the multivariate logistic regression analysis, NIHSS score on admission (adjusted OR: 1.23, 95% C.I.: 1.20–1.26, p = 0.001) and febrile on admission (adjusted OR: 3.65, 95% C.I. 1.36–9.83, p = 0.01), were found to be associated with poor outcome at 90 days (Table 2 and Fig 2). There was no statistical association for BMI, prior hypertension, ICU admission and intubated during admission. The regression model was able to discriminate 83% accurately for the 90 days poor outcome.

**Table 2. Multivariate analysis of the variables associated with 90-day poor outcome in all three groups.**

| VARIABLE | ODDS RATIO | 95% CI | | P Value |
|---|---|---|---|---|
| | | LOWER | UPPER | |
| Age | 1.06 | 1.04 | 1.07 | <0.001 |
| Sex | 0.55 | 0.37 | 0.81 | 0.002 |
| BMI | 1.00 | 0.97 | 1.03 | 0.86 |
| Prior Hypertension | 1.14 | 0.75 | 1.73 | 0.53 |
| Prior Dyslipidemia | 1.42 | 1.03 | 1.96 | 0.04 |
| Prior Stroke | 1.08 | 0.69 | 1.68 | 0.72 |
| Febrile on Admission | 6.02 | 1.69 | 21.38 | 0.006 |
| NIHSS score on Admission | 1.19 | 1.15 | 1.23 | <0.001 |
| ICU Admission | 1.17 | 0.56 | 2.49 | 0.69 |
| Systolic Blood Pressure | 1.0 | 0.99 | 1.01 | 0.85 |
| Platelet count | 1.00 | 0.99 | 1.00 | 0.17 |

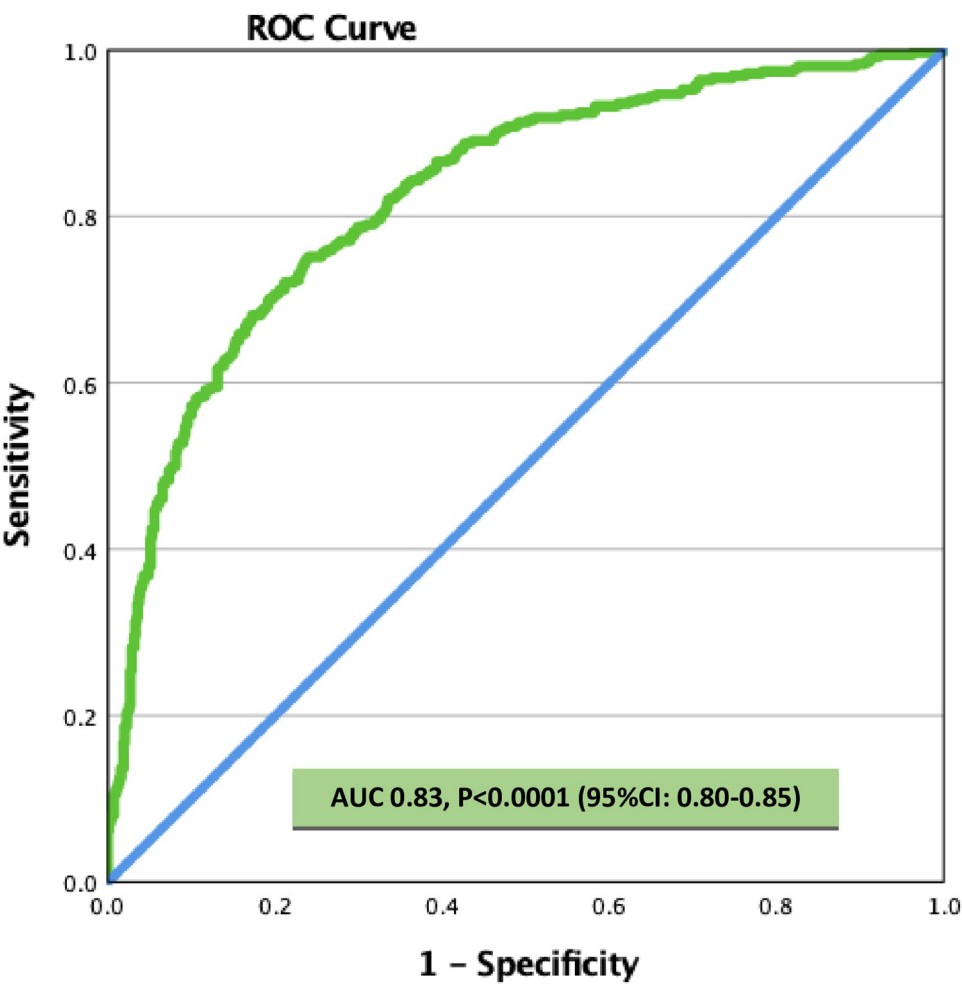

**Fig 2. ROC CURVE- To predict accuracy of 90-day poor outcome from the model.**

## Discussion

Patients with active COVID-19 related stroke had severe stroke and were also more likely to be febrile, requiring intubation and ICU admissions, and longer hospital stay. The most important new observation from our study relates to the stroke in patients with full recovery following COVID-19 infection. These patients had a rt-PCR confirmed diagnosis of COVID-19 infection and the stroke occurred following weeks to months of complete recovery of the viral infection. The overall pattern of stroke in this group was similar to and the profile and clinical course of patients with stroke prior to the pandemic. The stroke was likely related to the underlying vascular risk factors and not due to possible long-term sequala of the COVID-19 infection although we cannot be entirely certain of any potential relationship.

It is important to note that all patients in the post-C group had fully recovered from the viral infection and none had symptoms commonly associated with COVID-19 long-haulers. When comparing to the 1324 patients who had a stroke prior to the COVID-19 pandemic, the post-C had identical presentation, risk factors, clinical course, and prognosis. It is also interesting that once the patients recovered, the types of stroke as defined by the TOAST criteria [11] were very similar to what we had observed over in patients in the 14 months prior to COVID-19 pandemic.

Our study suggests that COVID-19 did not contribute to the etiology of stroke once the patient recovers. There are however several factors related to COVID-19 that may increase the risk of stroke in patients who have recovered and these needs attention [13, 14]. Potential mechanisms include continued endothelial injury [13], cardioembolism and potential paradoxical embolism via a PFO [15] or arterial dissection [16]. While the recovery is complete following COVID-19 in most patients, the "long-haulers" may have a prolonged inflammatory and prothrombotic state and therefore at a high risk for complications [17]. COVID-19 infection results in injury to the arterial endothelium, resulting in a prothrombotic state [13]. The prothrombotic state may persist and increase the risk of stroke. Cardiac muscle injury and heart failure seen with COVID-19 [15] may potentially contribute to embolic stroke in some cases. Cardioembolism was the final diagnosis in 14% of our patients with stroke following recovery from COVID-19 which is lower than the 25% seen in pre-COVID-19 cases and therefore likely did not contribute to the post-COVID-19 cases. Similarly, there were no cases of arterial dissection in the post-COVID-19 group.

There are strengths to our study. The Qatar Stroke Database is very robust and has prospectively recorded stroke trends in the country for more than 7 years. While the prospective data collection had shown a steady increase in admission rates over several years, the dramatic decline during over three months as the number of COVID-19 cases is very striking [8]. This is similar to multiple observations from around the world as noted in a recent meta-analysis from our group [18]. Our study shows that active COVID-19 positive stroke patients were more likely to be sicker, had more cortical involvement and had prolonged LOC and fewer frequency of good recovery at discharge. We also showed that patients who suffer a stroke following recovery from COVID-19 has similar characteristics to pre-COVID-19 cases.

The study has some limitations. A change over three months is brief and may not be sufficient to completely understand COVID-19-related changes. We noted higher rates of fever in the post-COVID patients. Although we are confident that none of the patients had active COVID-19, we cannot rule this out with certainty, nor can we rule out the possibility that this group of patients were on a higher risk for other infections. We did not document the relationship between the severity of COVID-19 and stroke. We also do not have enough long-term follow-up data at present on the patients seen during the pandemic to adequately document the changes in outcomes.

In summary, we present a comparison study on stroke subtypes prior to the pandemic to COVID-19 positive cases, and stroke in patients who recovered from the illness. Our data in 43 patients who had recovered from COVID-19 is reassuring in indicating no short-term effects of the illness.

## Acknowledgments

We acknowledge the assistance of all involved physicians, nurses, and staff of the Stroke Team in HMC. We also thank Ms. Reny Francis (HMC) and Kath McKenzie (University of Alberta) for her editorial assistance and supportive care.

## Author Contributions

**Conceptualization:** Naveed Akhtar, Fatma Abid, Ashfaq Shuaib.

**Data curation:** Yahia Imam, Salman Al-Jerdi, Sarah Salamah, Rand Al Attar, Muhammad Yasir, Hammad Shabir, Deborah Morgan, Sujatha Joseph.

**Formal analysis:** Rajvir Singh.

**Methodology:** Naveed Akhtar, Saadat Kamran, Salman Al-Jerdi.

**Project administration:** Muna AlMaslamani.

**Software:** Rajvir Singh.

**Supervision:** Naveed Akhtar, Ashfaq Shuaib.

**Validation:** Fatma Abid.

**Writing – original draft:** Naveed Akhtar, Ashfaq Shuaib.

**Writing – review & editing:** Fatma Abid, Saadat Kamran, Yahia Imam, Salman Al-Jerdi, Sarah Salamah, Rand Al Attar, Muhammad Yasir, Muna AlMaslamani, Ashfaq Shuaib.

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
