## [Decision Letter · Decision Letter 0]

23 Feb 2022

PONE-D-21-32603Characteristics and comparisons of acute stroke in “recovered" to “active COVID-19 and “pre-pandemic” in Qatar database.PLOS ONE

Dear Dr. Shuaib,

Thank you for submitting your manuscript to PLOS ONE. After careful consideration, we feel that it has merit but does not fully meet PLOS ONE’s publication criteria as it currently stands. Therefore, we invite you to submit a revised version of the manuscript that addresses the points raised during the review process.

Editor's comments:

1. Please follow the PLoS One's guideline for manuscript preparation.

2. Please cite (at least) five relevant articles from PLoS One.

3. Please revise the manuscript per the reviewers' comments.

We look forward to receiving your revised manuscript.

Kind regards,

Farzad Taghizadeh-Hesary

Academic Editor

PLOS ONE

Journal Requirements:

Reviewer 1:

The manuscript must describe a technically sound piece of scientific research with data that supports the conclusions. The authors made all data underlying the findings in their manuscript fully available without restriction. However there are some issues about the language and grammar.

to authors: in the abstract the period of the study is between Jan 2020 and Feb 2021, while in the methods section in the manuscript you did not mention that only you mention that patients between January-2019 to February-2020 were evaluated for the study (PPS) and served as the reference comparator for the COVID-19 cases. # At the end of introduction (We also evaluated the characteristics of stroke in post-COVID-19 stroke to no history of COVID-19 infection) please consider revising language. # At results :male/female 2404 (73.7%)/4860 (26.3%)] please correct number. Why did not you exclude the stroke mimic patients from the clinical characteristics and risk factors studied? it would differ in the percentage of each studied parameter. # in discussion: (None of our patients had symptoms associated with long haulers and as best as best as we can determine) please correct. # (This is similar to our recent meta-analysis of multiple observations from around the world (ADD our recent meta-analysis) please put the reference.

#( A change over three months is brief and may not be sufficient to completely understand COVID-19-related changes)this statement is not clear what do you mean by it?

Reviewer 2:

The timing of the manuscript is perfect given the ongoing pandemic. This study has provided new knowledge on stroke patients who have recovered from COVID-19. This is in addition to the already published data on acute stroke with acute Covid-19 infection.

However, there are a few minor comments

1) The authors should provide the population the HGH provides stroke services to since the hospital admits 95% of strokes in Qatar.

2) Statistics:

a) For consistency, p value should be added to “more frequently required intubation (CS:31.3%, PPS: 5.1% and RCS: 3.2%)” similar to the ICU admissions.

b) P value for NIHSS, ICU admission and other three-group comparisons (p. 16-18 of PDF) – a single p value is reported, presumably reflecting the ANOVA; the post-hoc Bonferroni p values should also be reported, at least in supplementary

3) The authors should remove or edit the comment “ ADD our recent meta-analysis” in the sentence “This is similar to our recent meta-analysis of multiple observations from around the world (ADD our recent meta-analysis) ( page 19 of PDF)

Reviewer 3:

The objective of this article was to compare characteristics and outcomes among acute stroke patients prior to the COVID-19 pandemic, stroke patients during active COVID-19 infection, and stroke patients after recovering from COVID-19. I appreciate the authors' revision of the manuscript. I believe this is an interesting article that would be of interest to the journals’ readers, but a number of improvements are still needed:

ABSTRACT

1. Some of the terminology should be more specific/clarified in the results section. For example: (1) What specific outcome do you mean by “poor stroke recovery”?, (2) “Large vessel disease and cardioembolic disease was significantly higher…”. Do you mean these were significantly more common?, (3) “There was a significant decline in stroke mimics…” Do you mean stroke mimics were less common, and if so, compared to which groups?

2. Methods: You mention that you included stroke cases between January 2020 and Feb 2021. However, in the full methods section, you stated that you included stroke patients admitted to HGH starting from January 2019. Please clarify

INTRODUCTION

1. “The number of COVID-19 worldwide…”. You should specify that this is the number of COVID-19 cases

2. While I appreciate the added detail in the introduction, make sure that everything is relevant to your study and it is clear why it is relevant for the reader. For example, what is the relevance of your prior publications that stroke occurs at a young age and acute stroke in COVID-19 pandemic from Qatar?

3. It doesn’t appear that the main objective was to evaluate the RISK of stroke (ie, the risk of having a stroke among all individuals in the source population who recovered from COVID-19- there is no quantification of this anywhere). Be careful with terminology.

METHODS

1. How did you define RCS patients? This should be clearly stated, because you need to define your main exposure group. Was it acute stroke within a specific number of days of recovering from COVID-19 infection? Would someone who had a stroke 1 year after recovering from COVID-19 be considered in the same group as individuals who had a stroke 1 month after recovering?

2. Statistical analysis: In your multivariate logistic regression model, did you also include a variable for the category of stroke (COVID-19 stroke, pre-COVID-19 stroke, recovered stroke)? Or was this model only run on patients with recovered stroke?

RESULTS

1. Under “demographic characteristics in the three groups”, the second and third sentences do not belong in the results section and should be moved to the discussion (interpretation of results should be limited to the discussion section only)

2. Same for the discussion about stroke mimics (“The details and possible reasons for the decrease in the percentage of stroke-mimics…”.. “Small vessel disease (SVD)…”)

TABLE

1. Table 1- Should clarify that BMI is mean +/- standard deviation

Finally, please review the article thoroughly for typos.

Reviewers' comments:

Reviewer's Responses to Questions

**Comments to the Author**

1. Is the manuscript technically sound, and do the data support the conclusions?

Reviewer #1: Yes

Reviewer #2: Yes

Reviewer #3: Yes

2. Has the statistical analysis been performed appropriately and rigorously? 

Reviewer #1: N/A

Reviewer #2: Yes

Reviewer #3: Yes

3. Have the authors made all data underlying the findings in their manuscript fully available?

Reviewer #1: Yes

Reviewer #2: Yes

Reviewer #3: No

4. Is the manuscript presented in an intelligible fashion and written in standard English?

Reviewer #1: Yes

Reviewer #2: Yes

Reviewer #3: No

5. Review Comments to the Author

Reviewer #1: The manuscript must describe a technically sound piece of scientific research with data that supports the conclusions. The authors made all data underlying the findings in their manuscript fully available without restriction. However there are some issues about the language and grammar.

to authors: in the abstract the period of the study is between Jan 2020 and Feb 2021, while in the methods section in the manuscript you did not mention that only you mention that patients between January-2019 to February-2020 were

evaluated for the study (PPS) and served as the reference comparator for the COVID-19 cases. # At the end of introduction (We also evaluated the characteristics of stroke in post-COVID-19 stroke to no history of COVID-19 infection) please consider revising language. # At results :male/female 2404 (73.7%)/4860 (26.3%)] please correct number. Why did not you exclude the stroke mimic patients from the clinical characteristics and risk factors studied? it would differ in the percentage of each studied parameter. # in discussion: (None of our patients had symptoms associated with long haulers and as best as best as we can determine) please correct. # (This is similar to our recent meta-analysis of multiple observations from around the world (ADD our recent meta-analysis) please put the reference.

#( A change over three months is brief and may not be sufficient to completely understand COVID-19-related changes)this statement is not clear what do you mean by it?

Reviewer #2: The timing of the manuscript is perfect given the ongoing pandemic. This study has provided new knowledge on stroke patients who have recovered from COVID-19. This is in addition to the already published data on acute stroke with acute Covid-19 infection.

However, there are a few minor comments

1) The authors should provide the population the HGH provides stroke services to since the hospital admits 95% of strokes in Qatar.

2) Statistics:

a) For consistency, p value should be added to “more frequently required intubation (CS:31.3%, PPS: 5.1% and RCS: 3.2%)” similar to the ICU admissions.

b) P value for NIHSS, ICU admission and other three-group comparisons (p. 16-18 of PDF) – a single p value is reported, presumably reflecting the ANOVA; the post-hoc Bonferroni p values should also be reported, at least in supplementary

3) The authors should remove or edit the comment “ ADD our recent meta-analysis” in the sentence “This is similar to our recent meta-analysis of multiple observations from around the world (ADD our recent meta-analysis) ( page 19 of PDF)

Reviewer #3: The objective of this article was to compare characteristics and outcomes among acute stroke patients prior to the COVID-19 pandemic, stroke patients during active COVID-19 infection, and stroke patients after recovering from COVID-19. I appreciate the authors' revision of the manuscript. I believe this is an interesting article that would be of interest to the journals’ readers, but a number of improvements are still needed:

ABSTRACT

1. Some of the terminology should be more specific/clarified in the results section. For example: (1) What specific outcome do you mean by “poor stroke recovery”?, (2) “Large vessel disease and cardioembolic disease was significantly higher…”. Do you mean these were significantly more common?, (3) “There was a significant decline in stroke mimics…” Do you mean stroke mimics were less common, and if so, compared to which groups?

2. Methods: You mention that you included stroke cases between January 2020 and Feb 2021. However, in the full methods section, you stated that you included stroke patients admitted to HGH starting from January 2019. Please clarify

INTRODUCTION

1. “The number of COVID-19 worldwide…”. You should specify that this is the number of COVID-19 cases

2. While I appreciate the added detail in the introduction, make sure that everything is relevant to your study and it is clear why it is relevant for the reader. For example, what is the relevance of your prior publications that stroke occurs at a young age and acute stroke in COVID-19 pandemic from Qatar?

3. It doesn’t appear that the main objective was to evaluate the RISK of stroke (ie, the risk of having a stroke among all individuals in the source population who recovered from COVID-19- there is no quantification of this anywhere). Be careful with terminology.

METHODS

1. How did you define RCS patients? This should be clearly stated, because you need to define your main exposure group. Was it acute stroke within a specific number of days of recovering from COVID-19 infection? Would someone who had a stroke 1 year after recovering from COVID-19 be considered in the same group as individuals who had a stroke 1 month after recovering?

2. Statistical analysis: In your multivariate logistic regression model, did you also include a variable for the category of stroke (COVID-19 stroke, pre-COVID-19 stroke, recovered stroke)? Or was this model only run on patients with recovered stroke?

RESULTS

1. Under “demographic characteristics in the three groups”, the second and third sentences do not belong in the results section and should be moved to the discussion (interpretation of results should be limited to the discussion section only)

2. Same for the discussion about stroke mimics (“The details and possible reasons for the decrease in the percentage of stroke-mimics…”.. “Small vessel disease (SVD)…”)

TABLE

1. Table 1- Should clarify that BMI is mean +/- standard deviation

Finally, please review the article thoroughly for typos.

6. PLOS authors have the option to publish the peer review history of their article (what does this mean?). If published, this will include your full peer review and any attached files.

Reviewer #1: No

Reviewer #2: No

Reviewer #3: No

---

## [Author Response · Author response to Decision Letter 0]

16 Mar 2022

Reviewer 1:

The manuscript must describe a technically sound piece of scientific research with data that supports the conclusions. The authors made all data underlying the findings in their manuscript fully available without restriction. However there are some issues about the language and grammar.

to authors: in the abstract the period of the study is between Jan 2020 and Feb 2021, while in the methods section in the manuscript you did not mention that only you mention that patients between January-2019 to February-2020 were evaluated for the study (PPS) and served as the reference comparator for the COVID-19 cases. # At the end of introduction (We also evaluated the characteristics of stroke in post-COVID-19 stroke to no history of COVID-19 infection) please consider revising language. # At results: male/female 2404 (73.7%)/4860 (26.3%)] please correct number. Why did not you exclude the stroke mimic patients from the clinical characteristics and risk factors studied? it would differ in the percentage of each studied parameter. # in discussion: (None of our patients had symptoms associated with long haulers and as best as best as we can determine) please correct. # (This is similar to our recent meta-analysis of multiple observations from around the world (ADD our recent meta-analysis) please put the reference. #(A change over three months is brief and may not be sufficient to completely understand COVID-19-related changes)this statement is not clear what do you mean by it?

Response: We thank the reviewer for the comments. We have corrected the time period and have removed the “stroke mimics” and done a complete re-analysis of the data for the revised manuscript. We have revised the abstract to reflect the changes. Additional changes in the “Introduction” and “Discussion” section will, we hope, improve the quality of the research. 

Reviewer 2:

The timing of the manuscript is perfect given the ongoing pandemic. This study has provided new knowledge on stroke patients who have recovered from COVID-19. This is in addition to the already published data on acute stroke with acute Covid-19 infection.

However, there are a few minor comments

1) The authors should provide the population the HGH provides stroke services to since the hospital admits 95% of strokes in Qatar. 

We have added information in the “Methods” section on the catchment area of HGH.

2) Statistics:

a) For consistency, p value should be added to “more frequently required intubation (CS:31.3%, PPS: 5.1% and RCS: 3.2%)” similar to the ICU admissions.

We apologize for this error. This was a typing error. We have added the p values to the above-mentioned variables.

b) P value for NIHSS, ICU admission and other three-group comparisons (p. 16-18 of PDF) – a single p value is reported, presumably reflecting the ANOVA; the post-hoc Bonferroni p values should also be reported, at least in supplementary

We thank the reviewer for the comment. The post-hoc Bonferroni analysis for the continuous variables are added to the results section

We apologize for this error. We have added the p values to the 

3) The authors should remove or edit the comment “ ADD our recent meta-analysis” in the sentence “This is similar to our recent meta-analysis of multiple observations from around the world (ADD our recent meta-analysis) ( page 19 of PDF)

We apologize for the error. The correct manuscript/reference has been added

Reviewer 3:

The objective of this article was to compare characteristics and outcomes among acute stroke patients prior to the COVID-19 pandemic, stroke patients during active COVID-19 infection, and stroke patients after recovering from COVID-19. I appreciate the authors' revision of the manuscript. I believe this is an interesting article that would be of interest to the journals’ readers, but a number of improvements are still needed:

ABSTRACT

1. Some of the terminology should be more specific/clarified in the results section. For example: (1) What specific outcome do you mean by “poor stroke recovery”?, (2) “Large vessel disease and cardioembolic disease was significantly higher…”. Do you mean these were significantly more common?, (3) “There was a significant decline in stroke mimics…” Do you mean stroke mimics were less common, and if so, compared to which groups?

2. Methods: You mention that you included stroke cases between January 2020 and Feb 2021. However, in the full methods section, you stated that you included stroke patients admitted to HGH starting from January 2019. Please clarify

As noted in our response to reviewer 1, we have completely revised the abstract.

INTRODUCTION

1. “The number of COVID-19 worldwide…”. You should specify that this is the number of COVID-19 cases

2. While I appreciate the added detail in the introduction, make sure that everything is relevant to your study and it is clear why it is relevant for the reader. For example, what is the relevance of your prior publications that stroke occurs at a young age and acute stroke in COVID-19 pandemic from Qatar?

3. It doesn’t appear that the main objective was to evaluate the RISK of stroke (ie, the risk of having a stroke among all individuals in the source population who recovered from COVID-19- there is no quantification of this anywhere). Be careful with terminology.

We thank the reviewer for the comments. We have revised the “Introduction” section to make it more focused

METHODS

1. How did you define RCS patients? This should be clearly stated, because you need to define your main exposure group. Was it acute stroke within a specific number of days of recovering from COVID-19 infection? Would someone who had a stroke 1 year after recovering from COVID-19 be considered in the same group as individuals who had a stroke 1 month after recovering?

2. Statistical analysis: In your multivariate logistic regression model, did you also include a variable for the category of stroke (COVID-19 stroke, pre-COVID-19 stroke, recovered stroke)? Or was this model only run on patients with recovered stroke?

We thank the reviewer for the useful suggestion and we have made corrections accordingly

RESULTS

1. Under “demographic characteristics in the three groups”, the second and third sentences do not belong in the results section and should be moved to the discussion (interpretation of results should be limited to the discussion section only)

2. Same for the discussion about stroke mimics (“The details and possible reasons for the decrease in the percentage of stroke-mimics…”.. “Small vessel disease (SVD)…”)

We thank the reviewer for the comments and we have made appropriate changes

TABLE

1. Table 1- Should clarify that BMI is mean +/- standard deviation

We have changed it as suggested by the reviewer

Finally, please review the article thoroughly for typos.

Reviewers' comments:

Reviewer's Responses to Questions

Comments to the Author

1. Is the manuscript technically sound, and do the data support the conclusions?

Reviewer #1: Yes

Reviewer #2: Yes

Reviewer #3: Yes

2. Has the statistical analysis been performed appropriately and rigorously?

Reviewer #1: N/A

Reviewer #2: Yes

Reviewer #3: Yes

3. Have the authors made all data underlying the findings in their manuscript fully available?

Reviewer #1: Yes

Reviewer #2: Yes

Reviewer #3: No

4. Is the manuscript presented in an intelligible fashion and written in standard English?

Reviewer #1: Yes

Reviewer #2: Yes

Reviewer #3: No

5. Review Comments to the Author

---

## [Decision Letter · Decision Letter 1]

21 Apr 2022

PONE-D-21-32603R1Ischemic stroke in patients that recover from COVID-19: Comparisons to historical stroke prior to COVID-19 or stroke in patients with active COVID-19 infectionPLOS ONE

Dear Dr. Shuaib,

Thank you for submitting your manuscript to PLOS ONE. After careful consideration, we feel that it has merit but does not fully meet PLOS ONE’s publication criteria as it currently stands. Therefore, we invite you to submit a revised version of the manuscript that addresses the points raised during the review process.

- Please revise the manuscript per the Reviewers' 1 and 3 comments.

- The manuscript requires linguistic copy editing. It's certificate from a specialized language service is essential for the final approval.

We look forward to receiving your revised manuscript.

Kind regards,

Farzad Taghizadeh-Hesary

Academic Editor

PLOS ONE

Journal Requirements:

Reviewer 1:

the manuscript is technically sound and the data support the conclusions. However, the date of the study is not corrected in the abstract "We interrogated the Qatar stroke database for all stroke admissions between Jan 2020 and Feb 2021".

in the methods section you stated that "The diagnosis of the stroke was within four weeks after full clinical and imaging from COVID-19 recovery" for patients with post- covid stroke while in the results section you stated that "The mean duration of time between recovery from COVID-19 infection and stroke was 126.9±75.9 days (median 124 days)".

in discussion : "They were, however, more likely to be febrile on admission compared to pre-C" how can you explain fever in post -C stroke patients?

in discussion: " These patients had a rt-PCR confirmed diagnosis of COVID-17 infection" please correct.

"When comparing to the 1413 patients who had a stroke prior to the COVID-19 pandemic" please correct number of patients to 1324 for pre-covid stroke.

in discussion: "It is also interesting that once the patients recovered, the stroke subtypes were very similar to what we had observed over the past 7 years (8)." what do you mean by stroke subtype? and did you compare with stroke patients in the previous 7 years or only previous 14 months as you mention in methods.

at the end of discussion " Our data in 93 patients who had recovered from COVID-19 is reassuring in indicating no short-term effects of" please correct number of patients (43 not 93).

Reviewer 2:

The authors have addressed the comments and this has now improved the standard of the manuscript for publication. I look forward to reading the manuscript in its published form.

Reviewer 3:

Thank you for addressing the reviewers' comments in your revised manuscript. I have no further substantial comments. However, please do a final review for typographical errors (there was a typo of "COVID-17" in the manuscript).

Reviewers' comments:

Reviewer's Responses to Questions

**Comments to the Author**

1. If the authors have adequately addressed your comments raised in a previous round of review and you feel that this manuscript is now acceptable for publication, you may indicate that here to bypass the “Comments to the Author” section, enter your conflict of interest statement in the “Confidential to Editor” section, and submit your "Accept" recommendation.

Reviewer #1: All comments have been addressed

Reviewer #2: All comments have been addressed

Reviewer #3: All comments have been addressed

2. Is the manuscript technically sound, and do the data support the conclusions?

Reviewer #1: Yes

Reviewer #2: Yes

Reviewer #3: Yes

3. Has the statistical analysis been performed appropriately and rigorously? 

Reviewer #1: N/A

Reviewer #2: Yes

Reviewer #3: Yes

4. Have the authors made all data underlying the findings in their manuscript fully available?

Reviewer #1: Yes

Reviewer #2: Yes

Reviewer #3: Yes

5. Is the manuscript presented in an intelligible fashion and written in standard English?

Reviewer #1: Yes

Reviewer #2: Yes

Reviewer #3: No

6. Review Comments to the Author

Reviewer #1: the manuscript is technically sound and the data support the conclusions. However, the date of the study is not corrected in the abstract "We interrogated the Qatar stroke database for all stroke admissions between Jan 2020

and Feb 2021".

in the methods section you stated that "The diagnosis of the stroke was within four weeks after full clinical and imaging from COVID-19 recovery" for patients with post- covid stroke while in the results section you stated that "The mean duration of time between recovery from COVID-19 infection and stroke was 126.9±75.9 days (median 124 days)".

in discussion : "They were, however, more likely to be febrile on admission compared to pre-C" how can you explain fever in post -C stroke patients?

in discussion: " These patients had a rt-PCR confirmed diagnosis of COVID-17 infection" please correct.

"When comparing to the 1413 patients who had a stroke prior to the COVID-19 pandemic" please correct number of patients to 1324 for pre-covid stroke.

in discussion: "It is also interesting that once the patients recovered, the stroke subtypes were very similar to what we had observed over the past 7 years (8)." what do you mean by stroke subtype? and did you compare with stroke patients in the previous 7 years or only previous 14 months as you mention in methods.

at the end of discussion " Our data in 93 patients who had recovered from COVID-19 is reassuring in indicating no short-term effects of" please correct number of patients (43 not 93).

Reviewer #2: The authors have addressed the comments and this has now improved the standard of the manuscript for publication. I look forward to reading the manuscript in its published form.

Reviewer #3: Thank you for addressing the reviewers' comments in your revised manuscript. I have no further substantial comments. However, please do a final review for typographical errors (there was a typo of "COVID-17" in the manuscript).

7. PLOS authors have the option to publish the peer review history of their article (what does this mean?). If published, this will include your full peer review and any attached files.

Reviewer #1: No

Reviewer #2: No

Reviewer #3: No

---

## [Author Response · Author response to Decision Letter 1]

28 Apr 2022

We are thankful for the additional comments from reviewer #1. We have corrected the errors in the abstract and the discussion section (track changes). 

We have also explained the questions raised about the strokes subtypes. We appreciated the thoughtful comments.

---

## [Decision Letter · Decision Letter 2]

10 Jun 2022

Ischemic stroke in patients that recover from COVID-19: Comparisons to historical stroke prior to COVID-19 or stroke in patients with active COVID-19 infection

PONE-D-21-32603R2

Dear Dr. Shuaib,

We’re pleased to inform you that your manuscript has been judged scientifically suitable for publication and will be formally accepted for publication once it meets all outstanding technical requirements.

Kind regards,

Farzad Taghizadeh-Hesary

Academic Editor

PLOS ONE

Reviewers' comments:

Reviewer's Responses to Questions

**Comments to the Author**

1. If the authors have adequately addressed your comments raised in a previous round of review and you feel that this manuscript is now acceptable for publication, you may indicate that here to bypass the “Comments to the Author” section, enter your conflict of interest statement in the “Confidential to Editor” section, and submit your "Accept" recommendation.

Reviewer #1: All comments have been addressed

Reviewer #3: All comments have been addressed

2. Is the manuscript technically sound, and do the data support the conclusions?

Reviewer #1: Yes

Reviewer #3: Yes

3. Has the statistical analysis been performed appropriately and rigorously? 

Reviewer #1: Yes

Reviewer #3: Yes

4. Have the authors made all data underlying the findings in their manuscript fully available?

Reviewer #1: Yes

Reviewer #3: Yes

5. Is the manuscript presented in an intelligible fashion and written in standard English?

Reviewer #1: Yes

Reviewer #3: Yes

6. Review Comments to the Author

Reviewer #1: the manuscript is technically sound, and do the data support the conclusions. The authors have adequately addressed comments raised in a previous round of review and the manuscript is now acceptable for publication. The manuscript is presented in an intelligible fashion and written in standard English. The authors made all data underlying the findings in their manuscript fully available.

Reviewer #3: Thank you for addressing the reviewer comments. I believe this will make a valuable contribution to the research field.

7. PLOS authors have the option to publish the peer review history of their article (what does this mean?). If published, this will include your full peer review and any attached files.

Reviewer #1: No

Reviewer #3: No

---

## [Editor Report · Acceptance letter]

14 Jun 2022

PONE-D-21-32603R2 

Ischemic stroke in patients that recover from COVID-19: Comparisons to historical stroke prior to COVID-19 or stroke in patients with active COVID-19 infection 

Dear Dr. Shuaib:

I'm pleased to inform you that your manuscript has been deemed suitable for publication in PLOS ONE. Congratulations! Your manuscript is now with our production department. 

Kind regards, 

on behalf of

Dr. Farzad Taghizadeh-Hesary 

Academic Editor

PLOS ONE